# KVCompose: Efficient Structured KV Cache Compression with Composite Tokens

## Abstract

Large language models (LLMs) rely on key-value (KV) caches for efficient autoregressive decoding; however, cache size grows linearly with context length and model depth, becoming a major bottleneck in long-context inference. Prior KV cache compression methods either enforce rigid heuristics, disrupt tensor layouts with per-attention-head variability, or require specialized compute kernels.

We propose a simple, yet effective, KV cache compression framework based on attention-guided, layer-adaptive composite tokens. Our method aggregates attention scores to estimate token importance, selects head-specific tokens independently, and aligns them into composite tokens that respect the uniform cache structure required by existing inference engines. A global allocation mechanism further adapts retention budgets across layers, assigning more capacity to layers with informative tokens. This approach achieves significant memory reduction while preserving accuracy, consistently outperforming prior structured and semi-structured methods. Crucially, our approach remains fully compatible with standard inference pipelines, offering a practical and scalable solution for efficient long-context LLM deployment.

## 1 Introduction

Transformer-based large language models (LLMs) with extended context windows have achieved strong performance across diverse tasks ranging from long-document analysis to multi-turn, personalized conversational agents (Zhang et al., 2025; Li et al., 2025). By storing past tokens in the key-value (KV) cache, LLMs can attend to a large number of tokens without recomputation, enabling coherent processing over long horizons.

However, scaling context length imposes substantial computational and memory burdens (Feng et al., 2024). The cost of the attention mechanism increases linearly with context size during the *prefill stage* and quadratically in terms of memory required to store KV pairs for each attention head across all layers. Such costs limit practical deployment for high-throughput or latency-sensitive applications, especially in resource-constrained settings such as edge devices or batch-serving environments.

To mitigate these issues, KV cache compression has emerged as a promising direction to reduce storage and computational load while preserving model quality. Broadly, compression strategies can be classified into three families:

**Structured Eviction** methods remove entire tokens (or groups of tokens) across all heads and layers, maintaining dense and hardware-friendly tensor layouts. This ensures compatibility with existing inference engines such as vLLM (Kwon et al., 2023) or Huggingface's Transformers (Wolf et al., 2020), offering predictable speedups and reduced memory usage without custom kernel support. Among these methods, online eviction policies such as Token Omission Via Attention (TOVA) (Oren et al., 2024) remove the currently least-attended token once a bounded cache is filled (fixed budget), preserving fluency with minimal engineering but tying importance strictly to the next-token distribution at each step. StreamingLLM retains a small set of attention sink tokens (typically the first few tokens that consistently attract high attention) alongside the sliding-window KV cache. This anchors the attention computation and stabilizes the score distribution, thereby enabling LLMs trained with finite attention windows to operate on effectively unbounded text streams without finetuning.

Table 1: Operational comparison of KV cache methods.

| Method | Selection strategy | Streaming ready | No offline preprocessing | Eviction mechanism | Works without engine changes |
|---|---|---|---|---|---|
| **TOVA** | Token-wise, fixed budget | ✓ | ✓ | Structured | ✓ |
| **StreamingLLM** | Fixed policy | ✓ | ✓ | Structured | ✓ |
| **SnapKV** | Per-head, fixed budget | ✓ | ✓ | Structured | ✓ |
| **PyramidKV** | Token-wise, pyramid budget | ✓ | ✓ | Structured | ✓ |
| **DuoAttention** | Head-type, binary budget | ✓ | ✗ | Semi-structured | ✓ |
| **KVzip** | Per-head, adaptive budget | ✗ | ✓ | Unstructured | ✗ |
| **KVCompose (Ours)** | Composite, adaptive budget | ✓ | ✓ | Structured | ✓ |

Although simple and streaming-ready, these approaches remain limited in scenarios with varying token importance. In contrast, importance-based structured eviction ranks tokens using attention-derived statistics and retains those above a budget threshold. SnapKV (Li et al., 2024) selects a fixed number of tokens per attention head, retaining those with the highest estimated contribution to the attention distribution, which reduces memory usage while preserving influential context tokens. However, the uniform per-head budget and variability of retained subsets across heads complicate cache reuse and may limit practicality in deployment environments requiring consistent KV layouts. PyramidKV (Cai et al., 2024) partially addresses this limitation through a heuristic "pyramid schedule" that allocates larger budgets to early layers. Although more effective than uniform allocation, this approach relies on heuristic allocation rather than learned patterns.

**Semi-Structured Eviction** methods operate at an intermediate granularity, balancing flexibility with structural regularity. For example, DuoAttention (Xiao et al., 2024) introduces a binary budgeting scheme that separates heads into global and local categories: global heads retain longer contexts to capture long-range dependencies, while local heads are restricted to shorter windows. This design reduces the cache size while preserving essential dependencies, and it remains compatible with streaming scenarios. However, DuoAttention requires an optimization-based offline procedure to classify the heads using synthetic datasets, thus, introducing a computational overhead. In addition, its coarse granularity and reliance on stable head roles limit adaptability across tasks and domains. More generally, semi-structured methods capture redundancies beyond structured token eviction but typically induce additional computational burdens.

**Unstructured Eviction** methods zero-out KV entries for selected tokens or heads, producing sparse key/value matrices. Although potentially achieving high compression ratios, these approaches require specialized sparse-attention kernels and optimizations to realize practical acceleration, limiting applicability on commodity hardware. Within this family, Feng et al. (2024) demonstrate that allocating non-uniform token budgets across heads can yield improved compression efficiency. They propose *Ada*, a wrapper around structured methods that enforces such adaptive budgets; we refer to this class of wrapped methods as Ada-variants throughout the paper. KVzip (Kim et al., 2025) extends this line of work by adopting a reconstruction-based scoring mechanism, ranking KVs according to their contribution to context reconstruction fidelity. This produces a reusable, query-agnostic compressed cache with adaptive per-head budgets, making it particularly suitable for multi-query scenarios over a fixed context, albeit at the cost of higher upfront computation.

Table 1 summarizes the operational characteristics of representative methods. These approaches differ not only in trading-off accuracy and efficiency but also in terms of practical implementation, streaming readiness, cache reusability, need for offline preprocessing, and compatibility with standard inference engines such as vLLM or Huggingface's Transformers.

This work introduces *KVCompose*, a structured eviction strategy for KV cache compression that balances efficiency, accuracy, and deployability. Our main contributions are:

- **Attention-guided token scoring.** We propose a scoring mechanism that aggregates attention patterns over either task-aware or task-agnostic input distributions, enabling reliable estimation of token importance without requiring retraining or auxiliary models.

- **Composite tokens for structured eviction.** We introduce the notion of *composite tokens*, where each head independently composes new tokens by aggregating tokens originating from different positions across all heads according to their importance score (see Section 3.3). This design preserves the tensor structure required by standard inference engines while allowing finer-grained head-level token eviction.

- **Layer-adaptive budget allocation.** Instead of prescribing fixed per-layer schedules, we allocate a global retention budget across layers by ranking composite-token scores. This adaptive budgeting ensures that layers carrying more informative tokens receive larger capacity, improving robustness under compression.

Together, these contributions yield a compression method that is more selective than windowing or pyramid heuristics, lighter than reconstruction-based approaches, and fully compatible with existing inference pipelines. The proposed approach enables substantial KV cache reduction with minimal quality loss, making long-context LLM inference more efficient and accessible.

## 2 PRELIMINARY

### 2.1 TRANSFORMER ARCHITECTURE AND ATTENTION MECHANISM

LLMs are typically implemented as autoregressive multi-layer transformers with multi-head attention. Let $\mathcal{C} = \{t_1, t_2, \ldots, t_N\}$ denote an input context of $N$ tokens, with corresponding embeddings collected into $\mathbf{X} \in \mathbb{R}^{N \times d}$, where $d$ is the model dimension.

For each attention head $i \in 1, \ldots, H_{\mathrm{q}}$, the transformer[1] computes queries, keys, and values using learned linear projections $\mathbf{W}_i^Q, \mathbf{W}_i^K, \mathbf{W}_i^V \in \mathbb{R}^{d \times d_h}$, where $d_h$ is the head dimension:

$$\mathbf{Q}_i = \mathbf{X}\mathbf{W}_i^Q, \quad \mathbf{K}_i = \mathbf{X}\mathbf{W}_i^K, \quad \mathbf{V}_i = \mathbf{X}\mathbf{W}_i^V. \tag{1}$$

Then, during autoregressive generation, for any new token with embedding $\mathbf{x} \in \mathbb{R}^d$, the model computes the associated query, key, and value representations as follow:

$$\mathbf{q}_i = \mathbf{x}\mathbf{W}_i^Q, \quad \mathbf{k}_i = \mathbf{x}\mathbf{W}_i^K, \quad \mathbf{v}_i = \mathbf{x}\mathbf{W}_i^V. \tag{2}$$

The new query $\mathbf{q}_i$ attends to the cached keys $\mathbf{K}_i \in \mathbb{R}^{N \times d_h}$, producing an attention distribution over the context:

$$\mathbf{A}_i = \mathrm{softmax}\left(\frac{\mathbf{q}_i\mathbf{K}_i^\top}{\sqrt{d_h}}\right) \in \mathbb{R}^{1 \times N}. \tag{3}$$

The corresponding output is a weighted combination of cached values $\mathbf{V}_i$:

$$\mathbf{y} = \sum_{i=1}^{H_{\mathrm{q}}} \mathbf{A}_i\mathbf{V}_i\mathbf{W}_i^O, \tag{4}$$

where $\mathbf{W}_i^O \in \mathbb{R}^{d_h \times d}$ are the output projections for each head.

---

[1]In practice, for a model composed of $L$ layers, the query, key and value matrices $\mathbf{Q}_i^{(\ell)}, \mathbf{K}_i^{(\ell)}, \mathbf{V}_i^{(\ell)}$ are computed per layer $\ell = 1, \ldots, L$. For clarity, in this section, we remove the dependence on $\ell$ to keep the notation simple.

## 2.2 KV Cache and Grouped Query Attention

In practice, for each Transformer layer $\ell \in \{1, \dots, L\}$ and each attention head $i \in \{1, \dots, H_q\}$, the matrices $\mathbf{K}_i^{(\ell)}, \mathbf{V}_i^{(\ell)} \in \mathbb{R}^{N \times d_h}$ constitute the KV cache, which is incrementally constructed during decoding and reused at every step to avoid cumbersome re-computation over the full context. Since each new token contributes an additional key and value vector per head and per layer, the KV cache grows linearly with the context length as the total memory footprint scales as $\mathcal{O}(L \cdot H_q \cdot N \cdot d_h)$.

To reduce the memory footprint of the KV cache, modern LLMs use Grouped Query Attention (GQA) (Ainslie et al., 2023), which shares the keys and values between groups of query heads. Specifically, with this approach, the transformer uses only $H_{\mathrm{kv}}$ distinct key-value head pairs, where $H_{\mathrm{kv}} < H_q$. This creates groups of query heads, where each group is composed of $G = H_q / H_{\mathrm{kv}}$ query heads attending to the same KV head, thus reducing the memory footprint to $\mathcal{O}(L \cdot H_{\mathrm{kv}} \cdot N \cdot d_h)$.

Nevertheless, even with GQA, both memory consumption and memory-bandwidth overhead remain significant in long-context inference. For example, consider a model with $L = 32$ layers, $H_{\mathrm{kv}} = 8$ heads, and head dimension $d_h = 128$. At a context length of $N = 32000$, the KV cache requires

$$32 \times 8 \times 32000 \times 128 \times 2 \times 4 \text{ bytes} \simeq 8\text{GB}, \tag{5}$$

where the factor of 2 accounts for both keys and values and we assume 4 bytes per float (FP32). This illustrates how cache storage rapidly becomes the dominant resource bottleneck, motivating the need for KV cache compression techniques that further reduce storage and computation while preserving model quality.

## 3 Methodology

### 3.1 Problem Formulation

Let $\pi$ denote a Transformer model composed of $L$ layers and, for each layer $\ell \in \{1, \dots, L\}$, let $\mathbf{K}^{(\ell)}, \mathbf{V}^{(\ell)} \in \mathbb{R}^{H_{\mathrm{kv}} \times N \times d_h}$ denote the key and value tensors associated with a context $\mathcal{C}$ of length $N$. Also, we use $\mathcal{KV} = \{\mathbf{K}^{(\ell)}, \mathbf{V}^{(\ell)}\}_{\ell=1}^L$ to indicate the the full cache.

Our objective is to construct a **compressed cache** $\mathcal{KV}' = \{\mathbf{K}'^{(\ell)}, \mathbf{V}'^{(\ell)}\}_{\ell=1}^L$, such that the performance of $\pi$ operating with $\mathcal{KV}'$ does not deviate significantly from that achieved with $\mathcal{KV}$. To do so, we define the **cache compression ratio** as

$$r = 1 - \frac{|\mathcal{KV}'|}{|\mathcal{KV}|}, \quad r \in [0, 1], \tag{6}$$

where $|\cdot|$ denotes the total number of cache entries. In addition, let $\epsilon(\mathcal{KV}', \mathcal{KV})$ denote a bounded performance gap between the model using $\mathcal{KV}'$ and the model using $\mathcal{KV}$. The KV compression problem can then be formulated as follows:

$$\underset{\mathcal{KV}', \{N_\ell\}_{\ell=1}^L}{\text{maximize}} \quad r \tag{7}$$

$$\text{such that} \quad \epsilon(\mathcal{KV}', \mathcal{KV}) \leq \epsilon_0, \tag{8}$$

$$\mathbf{K}'^{(\ell)}, \mathbf{V}'^{(\ell)} \in \mathbb{R}^{H_{\mathrm{kv}} \times N_\ell \times d_h}, \quad \forall \ell = 1, \dots, L, \tag{9}$$

where $\epsilon_0$ is a user-specified tolerance on performance degradation, and $(N - N_\ell)$ corresponds to the number of tokens evicted at layer $\ell$.

This optimization captures three key requirements for practical KV compression:

1. **Memory Efficiency.** The objective 7 seeks to maximize $r$, ensuring significant memory reduction relative to the original cache.

2. **Model Performance Preservation.** Constraint 8 guarantees that the degradation in task performance remains within the tolerance $\epsilon_0$. In practice, $\epsilon$ is estimated over a representative evaluation set $\mathcal{T}$ of tasks:

$$\epsilon(\mathcal{KV}', \mathcal{KV}) = \frac{1}{|\mathcal{T}|} \sum_{\tau \in \mathcal{T}} \frac{R(\mathcal{KV}, \tau) - R(\mathcal{KV}', \tau)}{R(\mathcal{KV}, \tau)}, \tag{10}$$

where $R(\mathcal{KV}, \tau)$ denotes the reward or evaluation score of $\pi$ on task $\tau \in \mathcal{T}$ when using cache $\mathcal{KV}$.

3. **Computational Efficiency.** Constraint 9 enforces a *structured eviction strategy*, where each layer $\ell$ retains exactly $N_\ell$ tokens across all KV heads. This structure ensures that compression yields proportional reductions in both memory and compute cost during inference. In fact, in practice, KV caches are represented as unified tensors per layer, with keys and values stored across all heads. This tensorized representation requires the second dimension (sequence length) to be shared across heads. Allowing different numbers of retained tokens per head would therefore break this assumption and necessitate non-standard attention kernels (e.g., custom CUDA implementations). For practical deployability, we restrict attention to **uniform per-layer retention**, which maintains compatibility with existing inference engines. Eventually, $N_\ell$ varies across layers, allowing for layer-adaptive budgeting.

**Remark.** An alternative perspective is to minimize performance degradation given a target compression ratio $r_{\text{target}}$. In this case, the optimization problem can be expressed as follows:

$$\underset{\mathcal{KV}', \{N_\ell\}_{\ell=1}^L}{\text{minimize}} \quad \epsilon(\mathcal{KV}', \mathcal{KV}) \tag{11}$$

$$\text{s.t.} \quad |\mathcal{KV}'| \le (1 - r_{\text{target}})|\mathcal{KV}|, \tag{12}$$

$$\mathbf{K}'^{(\ell)}, \mathbf{V}'^{(\ell)} \in \mathbb{R}^{H_{\text{kv}} \times N_\ell \times d_h}, \quad \forall \ell = 1, \dots, L. \tag{13}$$

This formulation reflects scenarios where memory availability is fixed in advance (e.g., hardware constraints), and the objective is to allocate the cache budget in a way that minimizes accuracy loss.

In the following, we propose a novel approach for constructing compressed representations $\{\mathbf{K}'^{(\ell)}, \mathbf{V}'^{(\ell)}\}_{\ell=1}^L$, which dynamically determines the layer-specific retention sizes $\{N_\ell\}_{\ell=1}^L$. Our method adaptively allocates the cache budget across layers, thereby optimizing the trade-off between compression ratio and task performance preservation, while maintaining compatibility with standard inference pipelines.

## 3.2 ATTENTION-BASED TOKEN IMPORTANCE SCORING

Our approach leverages attention patterns to identify which context tokens are most critical to retain during KV cache compression. The central hypothesis is that tokens receiving consistently high attention across tasks are more influential for preserving model behavior, and therefore should be prioritized during retention. Accordingly, we estimate token importance by analyzing aggregated attention scores over a representative task set $\mathcal{T}$, which guides cache eviction decisions.

### 3.2.1 TASK SET CONSTRUCTION

The effectiveness of the proposed scoring procedure depends on the construction of the task set $\mathcal{T}$ used to estimate token importance. We consider three practical settings:

- **Task-aware.** $\mathcal{T}$ consists of known downstream tasks of interest. This setting corresponds to scenarios where compression is tailored to specific applications, enabling highly targeted importance estimation.

- **Task-agnostic.** $\mathcal{T} = \{\mathcal{C}\}$, where the model uses the context as the only source of the signal for importance estimation.

### 3.2.2 ATTENTION COLLECTION AND AGGREGATION

For each task $\tau \in \mathcal{T}$ containing $M_\tau$ tokens, we collect the attention weights measuring how much each task token attends to each context token. This yields a four-dimensional attention tensor $\mathbf{A} \in \mathbb{R}^{L \times H_q \times N \times M}$, where $M = \sum_{\tau \in \mathcal{T}} M_\tau$ is the total number of task tokens.

We first aggregate across all task tokens to obtain a per-context-token importance score for each layer and head:

$$\mathbf{S}_{\text{agg\_task}}^{(\ell, h, c)} = \text{Agg}_{\text{task}}\left(\{\mathbf{A}^{(\ell, h, c, m)}\}_{m=1}^M\right), \tag{14}$$

where $\text{Agg}(\cdot)$ denotes an aggregation operator (e.g., max-pooling, average-pooling).

To accommodate *Grouped Query Attention* (GQA), we note that multiple query heads share the same KV pairs. We therefore reshape $\mathbf{S}_{\text{agg\_task}} \in \mathbb{R}^{L \times H_q \times N}$ into $\mathbf{S}_{\text{agg\_task}} \in \mathbb{R}^{L \times H_{kv} \times G \times N}$, where $G = H_q / H_{kv}$ is the number of query heads per KV group. We then aggregate within each KV head across the groups:

$$\mathbf{S}_{\text{agg\_group}}^{(\ell,h,c)} = \text{Agg}_{\text{group}}\left(\{\mathbf{S}_{\text{agg\_task}}^{(\ell,g,c)}\}_{g=1}^{G}\right), \tag{15}$$

yielding per-token importance scores aligned with the KV grouping structure.

Finally, to promote mild consistency across heads for the same token, we augment each token score with the mean score across all heads:

$$\mathbf{S}^{(\ell,h,c)} = \mathbf{S}_{\text{agg\_group}}^{(\ell,h,c)} + \frac{1}{H_{kv}} \sum_{i=1}^{H_{kv}} \mathbf{S}_{\text{agg\_group}}^{(\ell,i,c)}. \tag{16}$$

The resulting $\mathbf{S}^{(\ell,h,c)}$ provides a structured, head- and layer-specific measure of context token importance that forms the basis for our adaptive KV cache compression.

### 3.3 LAYER-WISE COMPRESSION STRATEGY WITH COMPOSITE TOKENS

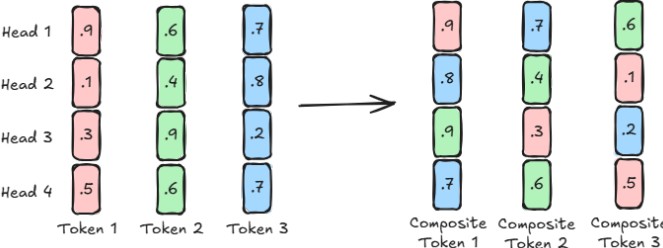

Figure 1: Composite token construction for layer-wise KV cache compression. Numbers represent importance scores. Each head independently selects its most important elements, creating composite tokens where positions may correspond to different original tokens across heads.

In practice, KV caches are stored as tensors where all heads within the same layer share a common sequence dimension. This tensorized representation enforces that the number of retained elements must be identical across heads in a given layer. However, a key observation follows from the self-attention formulation in equation 2–equation 4: although the number of retained elements must be uniform across heads, there is no requirement that these correspond to the *same* original context tokens across heads.

This insight enables a more flexible compression scheme through *composite token construction*. Rather than selecting a common subset of context tokens for all heads in a layer, we allow each head to independently select its most important elements, which are then aligned into shared positions (see Figure 1). We refer to these new tokens as the *composite tokens*.

Formally, for each layer $\ell$ and KV head $h$, we sort the original context positions by their importance scores:

$$\text{idx}^{(\ell,h)} = \text{argsort}\left(\{\mathbf{S}^{(\ell,h,c)}\}_{c=1}^{N}, \text{ descending=True}\right). \tag{17}$$

The $k$-th composite token at layer $\ell$ is then formed by aligning, across all heads, the token at position $\text{idx}^{(\ell,h)}[k]$ for head $h$. Thus, a composite token position may correspond to different original context tokens across heads. This new indexing allows us to define

$$\mathbf{S}'^{(\ell,h,k)} = \mathbf{S}^{(\ell,h,\text{idx}^{(\ell,h)}[k])}, \ \forall k = 1, \dots, N. \tag{18}$$

Then, to derive layer-wise retention budgets, we marginalize over heads to produce a composite importance score for each token position $k$:

$$\mathbf{I}^{(\ell,k)} = \text{Agg}_{\text{head}}\left(\left\{\mathbf{S}'^{(\ell,h,k)}\right\}_{h=1}^{H_{\text{kv}}}\right), \tag{19}$$

where $k \in \{1, \dots, N\}$ indexes composite token positions within layer $\ell$.

We then aggregate importance across all layers into a global pool:

$$\mathbf{P} = \left\{\mathbf{I}^{(\ell,k)} : \ell \in [1, L], \ k \in [1, N]\right\}. \tag{20}$$

Given a target compression ratio $r_{\text{target}}$, the total retention budget is

$$B_{\text{total}} = \left\lfloor (1 - r_{\text{target}}) \cdot L \cdot N \right\rfloor. \tag{21}$$

Then, for each layer $\ell$, we assign a budget

$$N_\ell = \left|\left\{k : \mathbf{I}^{(\ell,k)} \in \text{top-}B_{\text{total}} \text{ of } \mathbf{P}\right\}\right|, \tag{22}$$

allowing layers with more informative composite tokens to retain more elements, while less critical layers are compressed more aggressively.

Finally, for each layer $\ell$, we construct the compressed KV cache by selecting the top-$N_\ell$ elements for each head $h$:

$$\mathbf{K}'^{(\ell)}[h, :, :] = \mathbf{K}^{(\ell)}[h, \text{idx}^{(\ell,h)}[: N_\ell], :], \tag{23}$$

$$\mathbf{V}'^{(\ell)}[h, :, :] = \mathbf{V}^{(\ell)}[h, \text{idx}^{(\ell,h)}[: N_\ell], :]. \tag{24}$$

This process yields a compressed cache composed of *composite tokens*, where retained positions may originate from different context tokens across heads, while preserving the required tensor structure (see Figure 1). The approach conforms to the implementation constraints of existing inference engines while maximizing utilization of the compression budget, adaptively allocating capacity to the most informative layers and heads.

## 4 EXPERIMENTS

We evaluate our KV cache compression method on long-context reasoning tasks, comparing it against the aforementioned structured state-of-the-art eviction methods (see Table 1). We also report a simpler baseline method called ExpectedAttention, which prunes the KV cache based on precomputed expected attention weights during the generation phase (Jegou et al., 2024). DuoAttention is included as a baseline as well: although semi-structured, it does not require custom CUDA kernels to leverage the sparsity of the cache. Our experiments are designed to answer three key questions:

1. How well does our method preserve task accuracy under increasing compression ratios?
2. Does layer-adaptive composite token selection provide advantages over fixed or heuristic schedules?
3. How robust is the method across different models and evaluation setups (task-aware vs. task-agnostic)?

### 4.1 EVALUATION DATASET

We use the `Ruler-4096` (Hsieh et al., 2024) dataset from the `kvpress` benchmark (Jegou et al., 2024), which contains 6500 context–question pairs, each with a context length of up to 4096 tokens. The dataset spans 13 task types grouped into 4 categories:

- **Needle in a haystack (NIAH) tasks** *(8 types)*: include single (S) and multi-key retrieval (MK), multi-value retrieval (MV), and multi-query retrieval (MQ), each requiring extraction of relevant key–value pairs from distractor-heavy contexts.
- **Aggregation tasks** *(2 types)*: include Common Word Extraction (CWE) and Frequent Word Extraction (FWE), which require identifying common or frequent words from noisy token distributions.

Table 2: Performance comparison of the maximum compression ratio achieved by different structured KV cache compression strategies under a predefined tolerance $\epsilon_0$.

| Compression methods | Task-agnostic | | | Task-aware | | | Avg. perf. |
|---|---|---|---|---|---|---|---|
| | Qwen2.5 7B-Instruct | Qwen3 14B | Llama-3.1 8B | Qwen2.5 7B-Instruct | Qwen3 14B | Llama-3.1 8B | |
| Tolerance $\epsilon_0 = 20\%$ | | | | | | | |
| KVCompose (ours) | **80.9** | **81.8** | **75.7** | 71.4 | **86.0** | **83.0** | **79.8** |
| TOVA | 46.9 | 57.7 | 49.0 | **80.8** | 50.9 | 81.4 | 61.1 |
| DuoAttention | 74.5 | 27.8 | 65.8 | 78.3 | 27.6 | 69.6 | 57.3 |
| SnapKV | 6.9 | 35.3 | 36.6 | 43.1 | 80.4 | 78.2 | 46.8 |
| PyramidKV | 6.0 | 26.9 | 31.6 | 26.7 | 80.4 | 78.2 | 41.7 |
| ExpectedAttention | 16.3 | 31.5 | 49.9 | 20.1 | 33.6 | 55.6 | 34.5 |
| StreamingLLM | 27.5 | 28.9 | 28.7 | 27.8 | 29.0 | 29.8 | 28.6 |
| Tolerance $\epsilon_0 = 10\%$ | | | | | | | |
| KVCompose (ours) | **75.5** | **78.1** | **67.7** | 57.8 | **73.8** | **67.5** | **70.1** |
| TOVA | 12.6 | 19.0 | 27.8 | 65.8 | 32.2 | 52.7 | 35.0 |
| DuoAttention | 72.7 | 23.7 | 63.6 | **74.9** | 23.6 | 65.5 | 54.0 |
| SnapKV | 3.5 | 25.4 | 14.1 | 14.1 | 70.5 | 57.3 | 30.8 |
| PyramidKV | 3.0 | 17.7 | 8.5 | 9.7 | 46.2 | 47.6 | 22.1 |
| ExpectedAttention | 8.5 | 15.0 | 30.7 | 9.5 | 15.7 | 31.8 | 18.5 |
| StreamingLLM | 14.3 | 15.3 | 15.4 | 14.7 | 15.2 | 16.0 | 15.2 |

- **Multi-hop tracing** *(1 type)*: Variable Tracking (VT), which follows chains of variable assignments to recover all names referring to the same value.

- **Question Answering (QA)** *(2 types)*: includes a standard SQuAD-style QA and a harder QA variant with stronger distractors or modified queries.

For each task $\tau \in \mathcal{T}$, the reward $R(\mathcal{KV}, \tau)$ in equation 10 is defined as the average accuracy over all associated questions. The accuracy for a single question is scored on a continuous scale between 0.0 and 1.0, reflecting both the completeness and correctness of the predicted answer relative to the ground truth. For readability, we report results as percentages by multiplying raw accuracy scores by 100. This evaluation protocol follows the standardized procedure of the `Ruler-4096` benchmark (Hsieh et al., 2024), to which we refer the reader for further implementation details.

## 4.2 EXPERIMENTAL SETUP

We evaluate three representative open-source LLMs with different architectures and model size: `LLaMA-3.1-8B`, `Qwen2.5-7B-Instruct` and `Qwen3-14B`. To illustrate the effect of cache compression, we present results using two complementary views:

- **Accuracy–compression curves.** For each model and dataset category, we plot accuracy as a function of the compression ratio $r$. These curves highlight how quickly accuracy degrades under compression and clearly show the robustness gap between our method and competing approaches, especially in high-compression regimes.

- **Aggregate AUC comparison.** To provide a concise summary across all compression ratios, we report the Area Under the Curve (AUC) for each method and model. This metric makes it easy to compare methods at a glance, consolidating results over multiple ratios into a single robustness metric.

In our experiments, $r$ varies in $\{0, 0.1, 0.25, 0.4, 0.5, 0.6, 0.7, 0.8, 0.9\}$. Moreover, we use *max-pooling* for $\text{Agg}_{\text{task}}$ in equation 14 and *average-pooling* for all other aggregation operations. In Appendix A.1, we explore how different aggregation choices affect performance; we also perform ablation studies on some key aspects of the presented method.

Table 3: Performance comparison of the AUC achieved by different structured KV cache compression strategies.

| Compression methods | Task-agnostic | | | Task-aware | | | Avg. perf. |
|---|---|---|---|---|---|---|---|
| | Qwen2.5 7B-Instruct | Qwen3 14B | Llama-3.1 8B | Qwen2.5 7B-Instruct | Qwen3 14B | Llama-3.1 8B | |
| KVCompose (ours) | **81.1** | **83.6** | **79.7** | 79.8 | **85.7** | **84.1** | **82.3** |
| TOVA | 65.7 | 68.9 | 70.8 | **82.4** | 70.2 | 82.2 | 73.4 |
| DuoAttention | 77.3 | 46.9 | 72.3 | 80.2 | 49.2 | 74.8 | 66.8 |
| SnapKV | 42.8 | 60.4 | 61.4 | 66.5 | 82.2 | 81.8 | 65.8 |
| PyramidKV | 37.2 | 54.2 | 59.8 | 58.4 | 78.8 | 80.9 | 61.5 |
| StreamingLLM | 58.5 | 59.4 | 57.4 | 58.4 | 59.1 | 59.0 | 58.6 |
| ExpectedAttention | 42.1 | 52.9 | 68.6 | 48.3 | 54.2 | 71.5 | 56.3 |

## 4.3 Key Results

We summarize the main findings of our evaluation across `Ruler-4096` tasks in Tables 2 and 3. Our results highlight three central outcomes:

**1. Superior Performance Under Tolerance Constraints.** Table 2 reports the achievable compression ratio given two error tolerances ($\epsilon_0 =10\%, 20\%$). Our method, *KVCompose*, consistently outperforms all baselines across models and setups. At $\epsilon_0 = 20\%$, KVCompose achieves an average performance of 79.8, exceeding the best competing method (TOVA, 61.1) by more than 18 points. Even under the stricter tolerance $\epsilon_0 = 10\%$, KVCompose sustains an average score of 70.1, far above the next-best method (54.0 with DuoAttention). This demonstrates that composite tokens and adaptive budgeting yield superior robustness, allowing aggressive compression while meeting predefined quality guarantees.

**2. Consistently Higher AUC Across Tasks.** Table 3 compares methods using the AUC metric, which summarizes robustness across all compression ratios. KVCompose achieves the highest AUC on five out of six evaluation settings and the best overall average (82.3). The improvements over strong structured baselines are substantial: +8.9 points over TOVA and +15.5 points over PyramidKV. These results confirm that KVCompose degrades more gracefully as compression increases, preserving accuracy even in high-compression regimes. More granular results can be found in A.3.

**3. Practical and Deployable Efficiency.** Beyond accuracy gains, KVCompose respects the tensorized cache structure required by existing inference engines, unlike methods that depend on per-head variability (e.g., SnapKV, KVzip). This ensures that the observed improvements translate directly into real memory and compute savings without requiring custom kernels or engine modifications. While our primary focus is on structured eviction methods, we also examine an unstructured variant of our approach and compare it against unstructured KV cache compression strategies. The results are reported in Appendix A.2.

## 5 Conclusion

We introduced *KVCompose*, a framework for KV cache compression that combines attention-guided token scoring, composite token construction, and adaptive layer-wise budgeting. Our method achieves state-of-the-art accuracy–compression trade-offs across diverse long-context tasks, consistently outperforming prior structured and semi-structured approaches, while remaining fully compatible with existing inference engines. Ablation studies show that both composite tokens and adaptive budgeting are key to sustaining performance under aggressive compression. Although unstructured variants can offer slightly higher accuracy, they require custom kernels and are impractical for deployment.

Overall, KVCompose provides an effective and deployable solution for scaling LLMs to longer contexts under realistic hardware constraints.

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

# A APPENDIX

In this section, we present additional experiments to further assess the performance of the proposed strategy for KV cache compression.

## A.1 ABLATION STUDIES

To better understand the contribution of each design component, we conduct ablation studies on `Llama3.1-8B` and `Qwen2.5-7B-Instruct` using 10% of data randomly sampled from the `Ruler-4096` benchmark.

### A.1.1 EFFECT OF AGGREGATION OPERATORS

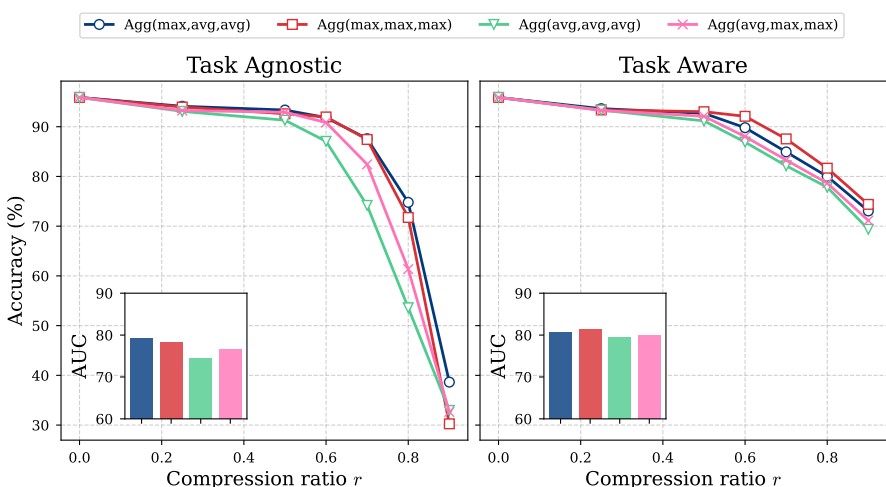

Figure 2: Effect of aggregation operators on the model accuracy in task-agnostic and task-aware scenarios. We use the label $\text{Agg}(x, y, z)$, $x, y, z \in \{\text{avg}, \text{max}\}$ to denote a configuration where $\text{Agg}_{\text{task}} = x$, $\text{Agg}_{\text{group}} = y$, and $\text{Agg}_{\text{head}} = z$. Results are reported for Llama-3.1 8B.

In Figure 2, we report results comparing different operator choices, i.e., max-pooling and average-pooling, for $\text{Agg}_{\text{task}}$, $\text{Agg}_{\text{group}}$, and $\text{Agg}_{\text{head}}$ To maintain a good level of readability, we report in Figure 2 the most significant operator combinations. Max-pooling for $\text{Agg}_{\text{task}}$ consistently yields stronger performance, indicating that preserving the strongest attention signal is most informative for token importance estimation. Keeping this setting, we also observed that for group and head aggregation, average-pooling provides the best accuracy for task-agnostic scenarios, while max-pooling leads to slight improvements in task-aware scenarios.

### A.1.2 EFFECT OF THE MEAN ON THE TOKEN SCORES

In this appendix, we assess the benefit of augmenting token scores with their mean across all heads (see equation 16). Figure 3 shows that this augmentation (W/ mean) improves performance across compression ratios, in particular when using Qwen2.5-7B-Instruct in task-aware scenarios. Without the mean term (W/o mean), token importance estimates vary more strongly across heads, which can lead to inconsistent selections and accuracy loss under compression. Adding the mean provides a stabilizing signal, resulting in more reliable retention decisions and higher overall accuracy.

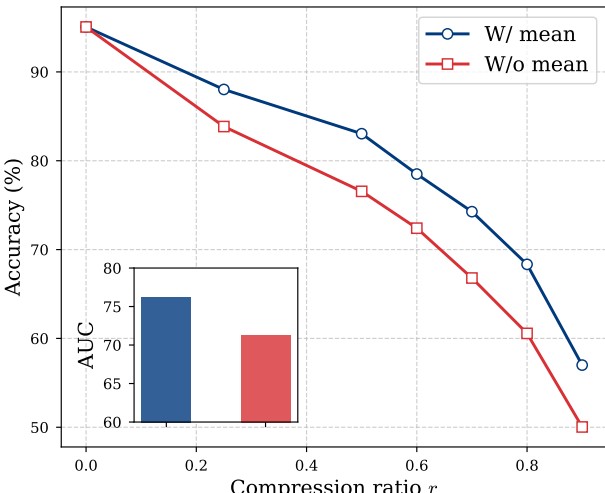

Figure 3: Effect of the mean on the token scores. Results are reported for Qwen2.5-7B-Instruct in task-aware setting.

### A.1.3 EFFECT OF VALUE NORMS ON TOKEN SCORES

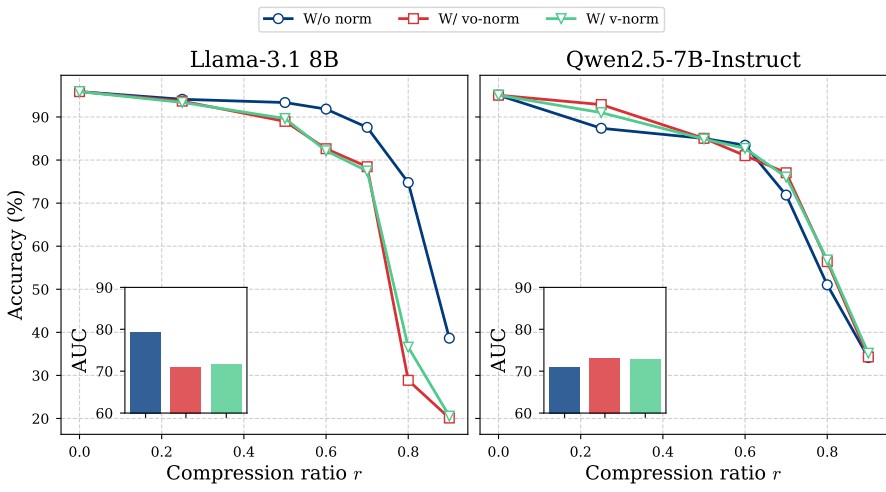

Figure 4: Effect of incorporating value norms into token scores on the model accuracy. Results are reported for task-agnostic settings.

ExpectedAttention (Jegou et al., 2024) has shown that scoring tokens based on attention weights only has some drawbacks, as this approach ignores the scale of the value vectors. Indeed, two tokens with identical attention weights may contribute very differently if their value vectors have significantly different norms. Recalling equation 4, the output of an attention layer can be rewritten

as:

$$\mathbf{y} = \sum_{i=1}^{H_q} \mathbf{A}_i \mathbf{V}_i \mathbf{W}_i^O \tag{25}$$

$$= \sum_{i=1}^{H_q} \left( \mathbf{A}_i \|\mathbf{V}_i\| \right) \frac{\mathbf{V}_i}{\|\mathbf{V}_i\|} \mathbf{W}_i^O \tag{26}$$

$$= \sum_{i=1}^{H_q} \left( \mathbf{A}_i \|\mathbf{V}_i \mathbf{W}_i^O\| \right) \frac{\mathbf{V}_i}{\|\mathbf{V}_i \mathbf{W}_i^O\|} \mathbf{W}_i^O. \tag{27}$$

Equations 26 and 27 show explicitly that the effective contribution of a token is scaled by its value norm (either before or after projection). Empirically, we observe that token value norms can vary by nearly two orders of magnitude (approximately $1$-$100\times$), indicating that norm differences are non-negligible in practice.

Motivated by this observation, we experiment with token scoring schemes that weight attention by the value norm. Specifically, we define a weighted attention matrix $\mathbf{A}_i^{(w)}$ in which each entry is multiplied by either the raw value norm $\|\mathbf{V}_i\|$ (*v-norm*) or the projected norm $\|\mathbf{V}_i \mathbf{W}_i^O\|$ (*vo-norm*). The *v-norm* variant corresponds to the approach used in ExpectedAttention (Jegou et al., 2024).

Our experiments reveal that the impact of the weighted attention matrix is model-dependent: we observed modest gains for Qwen models in some configurations, but a clear degradation for LLama-3.1 8B (see Figure 4). These results show that the attention-only criterion is a reliable and simple approach for our composite token scoring mechanism.

## A.2 DISCUSSION ON UNSTRUCTURED EVICTION METHODS

In this appendix, we evaluate an *unstructured variant* of our approach, obtained by removing the final aggregation step and allowing each head to select its tokens independently. For completeness, we compare the *unstructured variant* of our approach with *unstructured* KV cache compression strategies, summarized in Table 4. Unlike structured eviction, these approaches assign non-uniform budgets across heads and evict individual entries independently. While this often yields higher accuracy due to increased flexibility, such methods are not directly compatible with standard KV cache implementations and therefore require non-trivial engine modifications (Feng et al., 2024; Kim et al., 2025).

We include three representative unstructured baselines: (i) **KVzip** (Kim et al., 2025), which optimizes per-head adaptive budgets using a reconstruction-based criterion; (ii) **Ada-ExpectedAttention**; and (iii) **Ada-SnapKV**, both of which extend structured methods with the adaptive budget allocation mechanism proposed in Feng et al. (2024).

Evaluation relies on the *attention patching* utility of kvpress, which simulates per-head budget allocation by zeroing out keys with near-zero attention values. It is important to note that this procedure is only theoretical: it does not provide actual memory or runtime savings. Real efficiency gains would require custom CUDA kernels and careful adaptation of the model's attention implementation, as is the case for KVzip and Ada-variants.

Table 4: Performance comparison of unstructured KV cache compression strategies.

| Compression methods | Task-agnostic | | | |
| --- | --- | --- | --- | --- |
| | Qwen2.5 7B-Instruct | Qwen3 14B | Llama-3.1 8B | Avg. perf. |
| KVzip | **88.4** | **89.9** | **90.2** | **89.5** |
| Unstructured KVCompose (ours) | 87.3 | 89.1 | 88.9 | 88.4 |
| AdaExpectedAttention | 74.4 | 84.4 | 78.8 | 79.2 |
| AdaSnapKV | 49.5 | 64.5 | 67.2 | 60.4 |

Table 4 shows that, among the baselines, **KVzip** achieves the strongest results, with an average performance of 89.5, slightly outperforming our unstructured variant (88.4). Both substantially surpass the Ada-based methods, which remain below 80 on average. These results highlight two points: first, unstructured eviction indeed provides an upper bound on achievable accuracy under compression; second, our attention-guided scoring transfers effectively even in the unstructured setting, nearly matching KVzip despite being primarily designed for structured eviction.

Nevertheless, since structured methods are directly compatible with existing inference engines and yield actual memory and runtime savings, we focus our contributions on structured eviction. We view unstructured methods as valuable for understanding the potential performance bounds of KV cache compression, but less practical for deployment without significant engineering overhead.

### A.3 ADDITIONAL PERFORMANCE EVALUATION

In this appendix, we provide detailed results on the performance reported in Section 4.3. Specifically, Figure 5 presents a detailed comparison of the average performance of our method against state-of-the-art structured eviction strategies on the `Ruler-4096` dataset (Hsieh et al., 2024). Moreover, Figure 6-Figure 11 further breaks down the results of Figure 5 at the task level, providing a fine-grained view of performance across the 13 tasks in `Ruler-4096`.

**Remark:** PyramidKV extends SnapKV by assigning a predefined per-layer budget, giving larger budgets to lower layers. As the compression ratio increases, PyramidKV converges to SnapKV, which explains their identical performance in the high-compression regime ($r \geq 70\%$). In the mid-compression range ($r \approx 50\%$), however, the fixed budgeting scheme does not appear to align well with all models. This is particularly evident for `Qwen2.5-7B-Instruct` and `Qwen3-14B-Instruct`, where performance occasionally drops below that observed at higher compression ratios, producing an unusual curve shape.

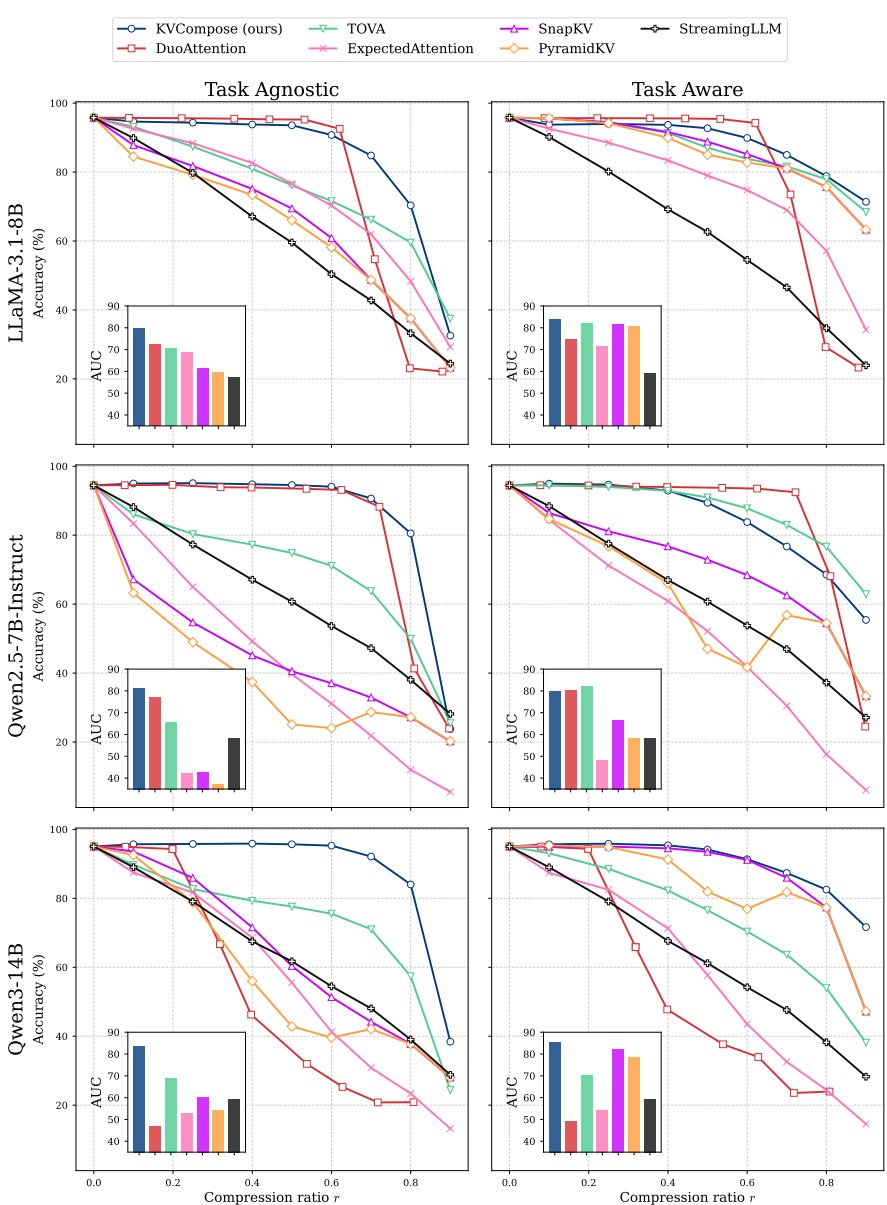

Figure 5: Performance comparison of different structured eviction methods for different compression ratios.

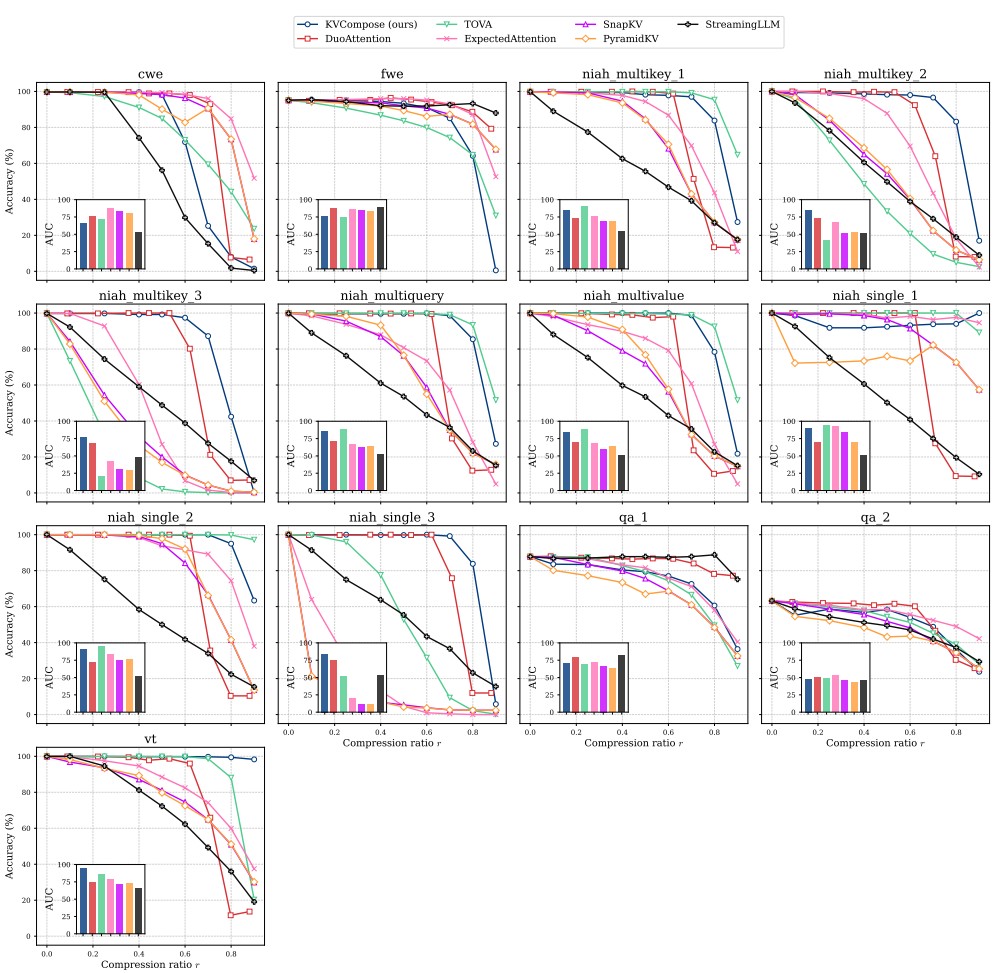

Figure 6: Performance comparison of the different eviction methods per sub-tasks of the Ruler dataset. Results are reported for `LLaMA-3.1-8B` in the task-agnostic setting.

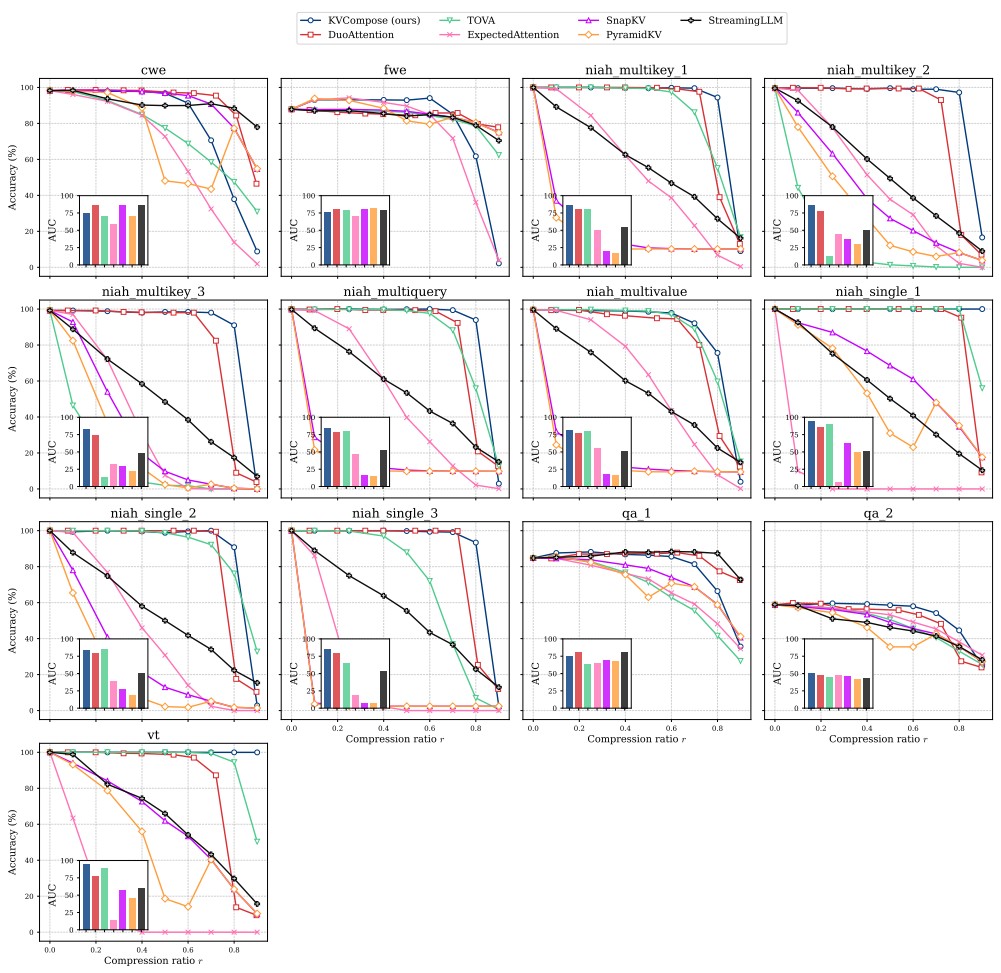

Figure 7: Performance comparison of the different eviction methods per sub-tasks of the Ruler dataset. Results are reported for `Qwen2.5-7B-Instruct` in the task-agnostic setting.

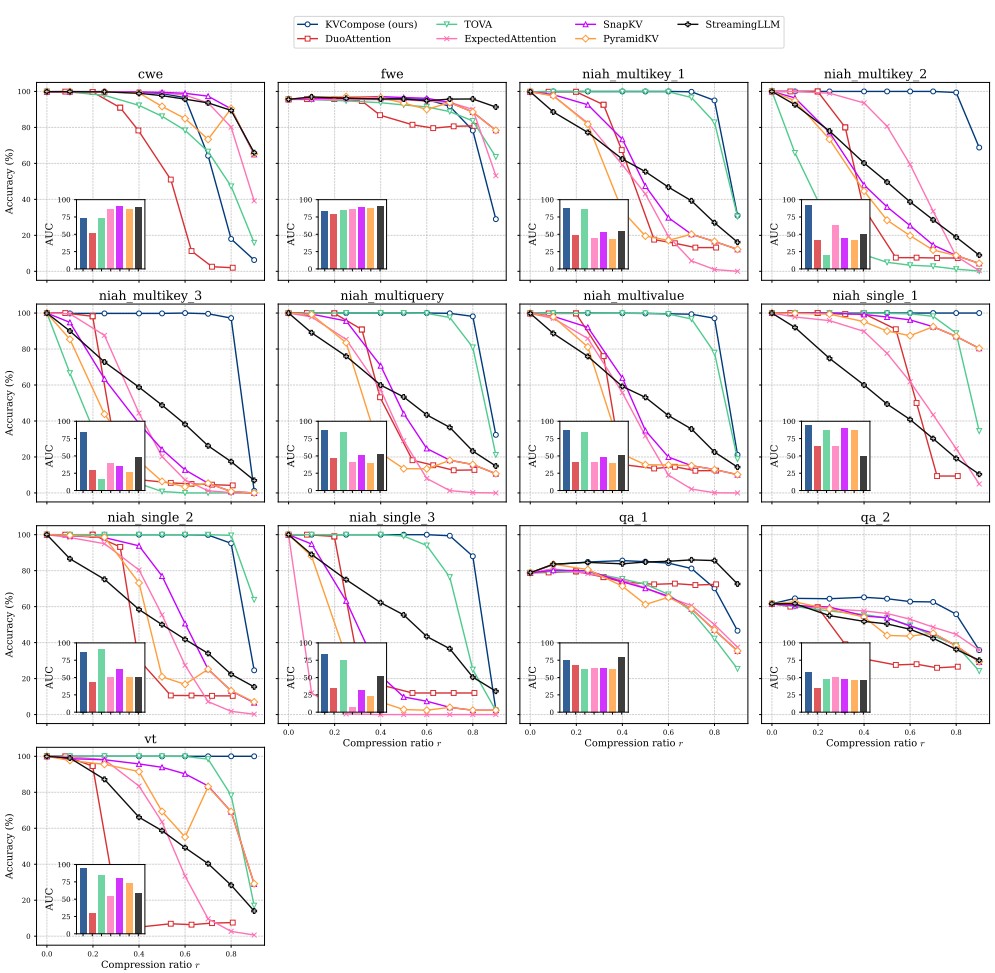

Figure 8: Performance comparison of the different eviction methods per sub-tasks of the Ruler dataset. Results are reported for `Qwen3-14B-Instruct` in the task-agnostic setting.

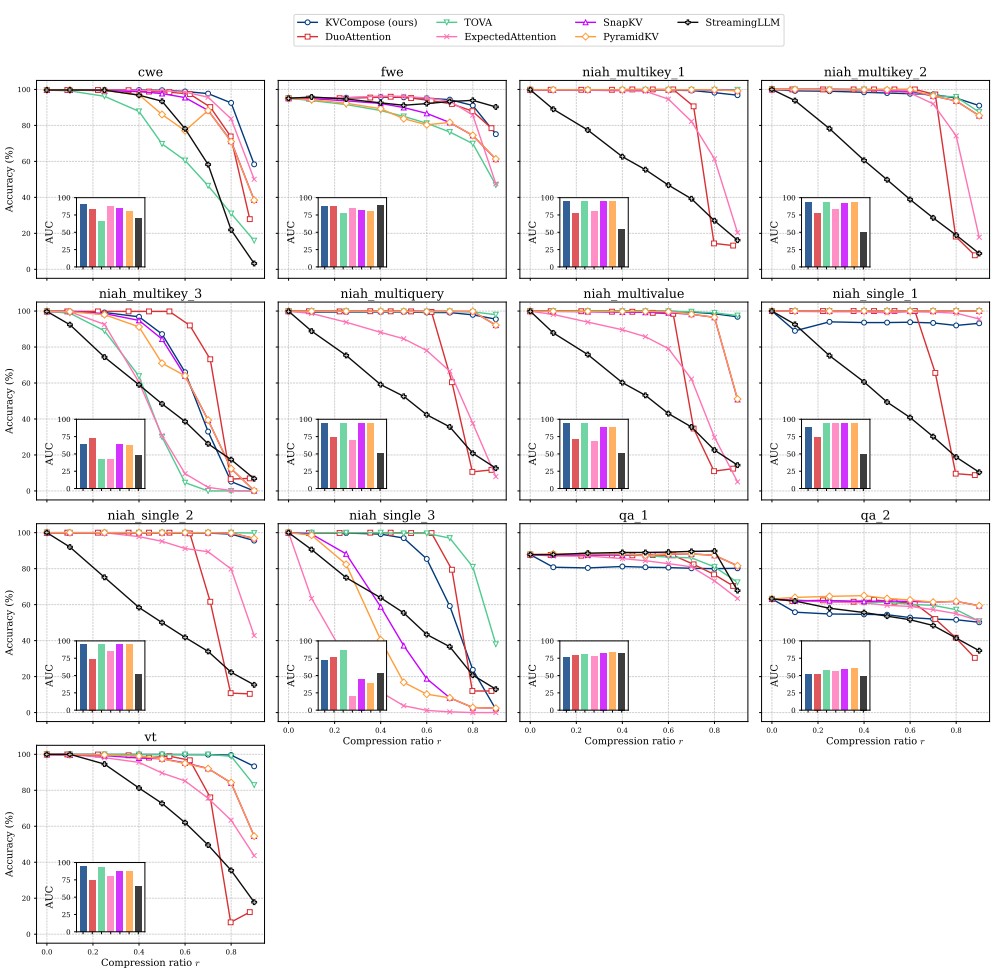

Figure 9: Performance comparison of the different eviction methods per sub-tasks of the Ruler dataset. Results are reported for `LLaMA-3.1-8B` in the task-aware setting.

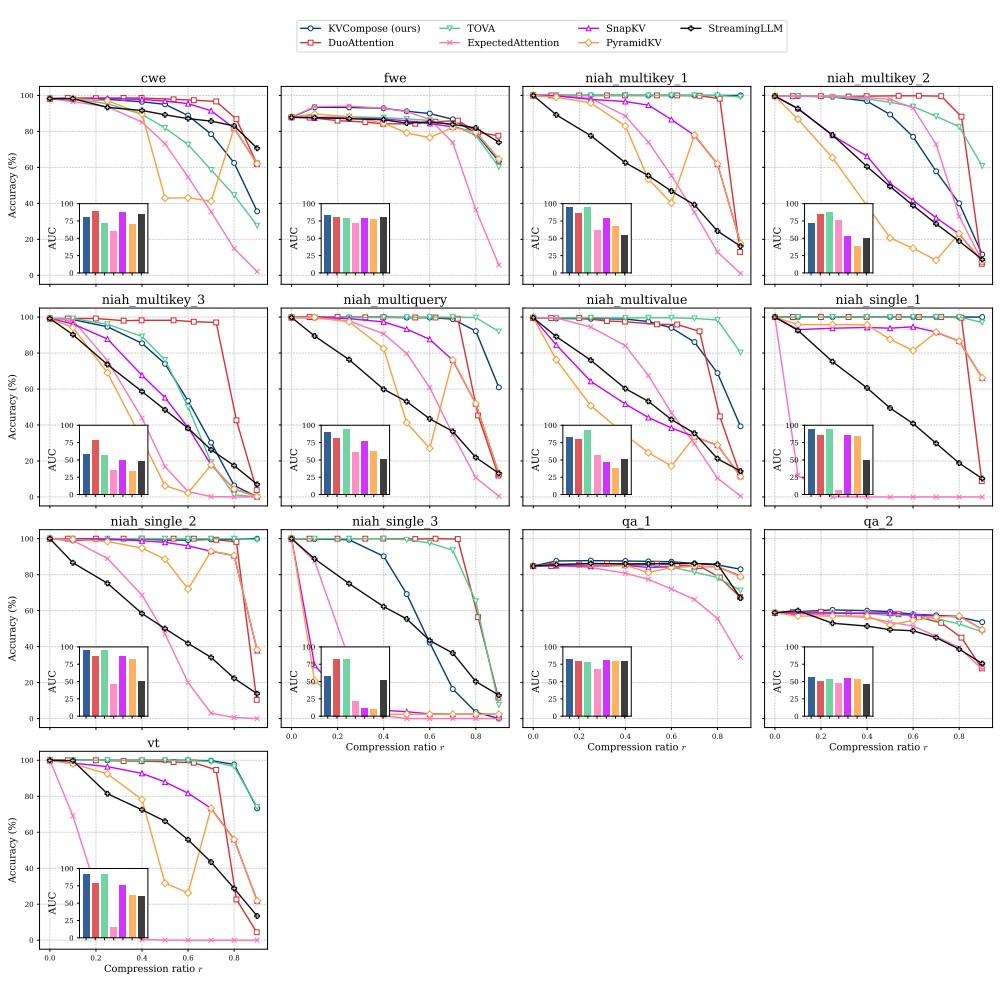

Figure 10: Performance comparison of the different eviction methods per sub-tasks of the Ruler dataset. Results are reported for `Qwen2.5-7B-Instruct` in the task-aware setting.

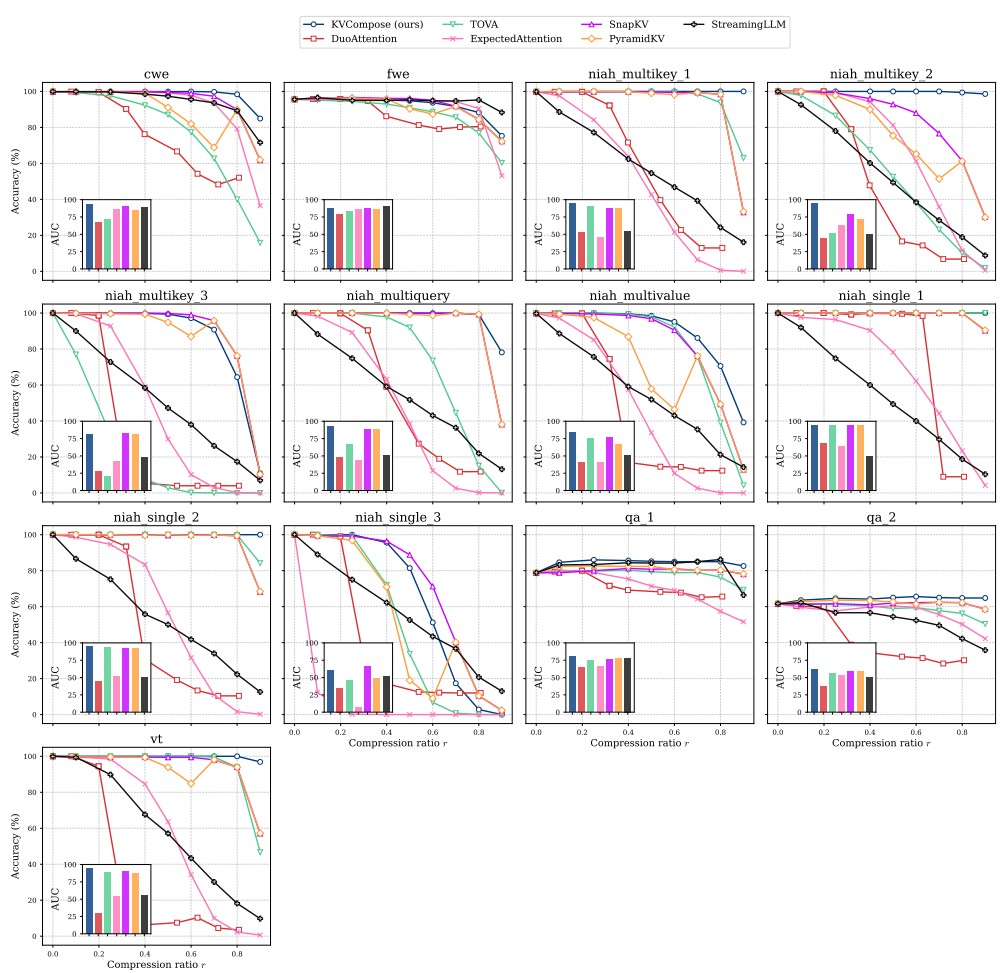

Figure 11: Performance comparison of the different eviction methods per sub-tasks of the Ruler dataset. Results are reported for `Qwen3-14B-Instruct` in the task-aware setting.

