# OpenReview forum: "KVCompose: Efficient Structured KV Cache Compression with Composite Tokens"
_ICLR.cc/2026/Conference — Submitted to ICLR 2026_

### Official Review · Reviewer_82QU · 2025-10-30

**Soundness:** 2
**Presentation:** 3
**Contribution:** 2
**Rating:** 6
**Confidence:** 3

**Summary:**

This paper presents KVCompose, a structured KV-cache compression method for LLM inference. The method first aggregates attention across tasks or contexts to score token importance and then creates composite tokens by allowing each KV head to select its own most important original positions, aligning those selections into a shared per-layer sequence dimension, which preserves the standard dense tensor format. A layer-adaptive global budget distributes retention across layers by ranking composite-token scores. Experiments on RULER-4096 with LLaMA-3.1-8B, Qwen2.5-7B-Instruct, and Qwen3-14B show higher AUC than other baselines such as TOVA, SnapKV, and PyramidKV.

**Strengths:**

* KVCompose is evaluated across various model families and model sizes.
* Ablation studies examine key design choices, such as operator selections for aggregation, Aggtask, Agggroup, and Agghead. These analyses help clarify the contribution of each component.

**Weaknesses:**

* No runtime latency and memory usage comparison. Reporting compression ratios alone may not fully reflect efficiency improvements. Latency and memory statistics would provide a clearer picture of the practical benefits of KVCompose.

* The range of benchmarks is somewhat limited. KVCompose has been evaluated only on RULER-4096. Evaluation with longer input lengths on RULER and on other widely used benchmarks, such as LongBench, would provide a more comprehensive assessment of KVCompose's performance.

* What is the computational cost of the token selection process? For example, it would be helpful to discuss whether KVCompose remains compatible with Flash-Attention, given that attention computation is explicitly involved. It may be useful to compare or discuss KVCompose in relation to recent structured eviction approaches, such as “LaCache: Ladder-Shaped KV Caching for Efficient Long-Context Modeling of Large Language Models” (ICML'25), particularly under accuracy-latency trade-offs.

**Questions:**

Please refer to the points raised in the Weaknesses section above.

---

> ### Author Response · Authors · 2025-12-02
>
> We thank Reviewer 4 for their positive feedback regarding our evaluation across model families and sizes, as well as their appreciation of our ablation studies examining aggregation operator choices. We address the remaining concerns below.
>
> **Runtime latency and memory usage (W1):**
> As discussed in the general response, KVCompose introduces computation during prefill, including attention aggregation, sorting of per-head importance scores, and indexed gathers to construct composite tokens. This overhead is non-negligible for single-query latency but is amortized in reusable cache scenarios, which are common in practice and where the one-time compression cost is offset by repeated inference savings. Importantly, KVCompose maintains a structured cache with uniform tensor layouts, allowing the compressed cache to be used directly with standard dense attention and with FlashAttention, without requiring custom CUDA kernels.
>
> **Benchmark scope (W2):**
> We refer the reviewer to the general response for a detailed explanation of our benchmark choices and computational constraints. Briefly, Ruler-4096 provides a standardized and diverse evaluation framework widely used in the KV cache compression literature. The full study requires over 1,500 GPU-hours across rates, models, methods, and settings. Following the reviewers’ recommendation, we extend our evaluation to Ruler-16k to validate performance at longer context lengths, focusing on LLaMA-3.1-8B due to time constraints.
>
> **Comparison with LaCache and streaming-oriented methods (W3):**
> We thank the reviewer for pointing us to LaCache. LaCache employs a ladder-shaped caching strategy with iterative compactification specifically designed for streaming long-context inference, where eviction decisions must be made online and per-token latency is critical. In contrast, KVCompose targets the reusable cache regime, where a compressed cache is computed once during prefill and reused for many subsequent queries. These are complementary use cases: streaming-oriented methods prioritize speed and rely on simpler, fixed rules, while KVCompose invests more computation upfront to achieve higher compression quality through attention-based scoring and composite tokens. Although streaming is not the primary focus of this work, KVCompose could be integrated into frameworks such as LaCache or RocketKV, where it could serve as a drop-in first-stage eviction mechanism. We will add a paragraph in the main text discussing these distinctions.

---

### Official Review · Reviewer_ZsMN · 2025-10-31

**Soundness:** 1
**Presentation:** 2
**Contribution:** 1
**Rating:** 2
**Confidence:** 4

**Summary:**

The paper introduces KVCompose. The core innovation is the concept of "composite tokens." Recognizing that while the number of tokens must be uniform across heads in a layer for tensor compatibility, the identity of those tokens need not be, KVCompose allows each head to independently select its most important tokens based on aggregated attention scores.  This aims to combine the flexibility of per-head eviction (unstructured) with the hardware efficiency of dense tensor layouts (structured).

**Strengths:**

The concept of "composite tokens" is an interesting attempt to bridge the gap between unstructured eviction flexibility and the hardware efficiency required by standard inference engines.

**Weaknesses:**

**Q1** The central claim is "fully compatible with standard inference pipelines" and "Works without engine changes". However, . Modern LLMs, including Llama and Qwen (used in the experiments), rely heavily on positional embeddings, often Rotary Positional Embeddings (RoPE). RoPE applies rotations based on the absolute position of tokens. KVCompose's core mechanism constructs composite tokens by aligning K/V entries from different original positions across different heads. For example, the k-th composite token might contain the K/V from position 50 for Head 1 and position 1000 for Head 2. Standard inference engines assume that all K/V entries at tensor index k share the same original position, applying a uniform positional embedding for that index. KVCompose does not follow this assumption.

**Q2** The claim that KVCompose is "Streaming ready" (Table 1) is misleading. The proposed scoring mechanism (Section 3.2) and global budget allocation (Eq. 22, 23) require processing the full context N to calculate attention statistics and determine budgets. This makes the approach suitable only for fixed-context compression (e.g., prompt compression during prefill) but not for true unbounded streaming applications (like StreamingLLM), which must make online eviction decisions as tokens arrive.

**Q3** I think the reorganization of the cache into composite tokens breaks the semantic and positional continuity of the cache structure. This fundamentally complicates or prevents the application of standard system optimizations crucial for efficient serving, such as prefix caching, cache sharing across requests, and speculative decoding.


**Q4** The evaluation is severely limited in context length, failing to validate the method's effectiveness for long-context inference. Experiments are solely conducted on Ruler-4096. This is insufficient, as KV bottlenecks become critical at much longer sequences. The method must be evaluated on significantly longer contexts (e.g., 32k, 128k+) with limited token budgets 256-512.

**Q5** The evaluation lacks essential metrics and task diversity. Performance on tasks like long-document summarization or complex QA (e.g., LongBench2). The impact of eviction on coherence is critical in multi-turn scenarios, requiring evaluation on benchmarks such as SCBench.

**Q6.** The experimental evaluation is weak as they are ignoring recent work such as Lcache, RocketKV, ShadowKV, and seerAttention.
At least you should compare with RocketKV and ShadowKV in different token budgets and show how far you are from them.

**Q7.** The paper claims efficiency but provides no empirical analysis of the computational overhead. KVCompose introduces significant computation during the prefill phase: attention aggregation, sorting scores for every head (O(L * H_kv * N log N)), and indexed gathers to reorganize the cache. The latency and throughput impact of these operations must be measured and reported.
Q8 The authors incorrectly state that the "memory required to store KV pairs" grows quadratically with context size. The KV cache memory footprint grows linearly (O(N)), as correctly noted later. The computational cost of prefill attention is quadratic (O(N^2)).
Q9. The memory calculation assumes FP32 (4 bytes). Modern deployments predominantly use FP16/BF16 (2 bytes); the calculation should reflect modern precision standards.

**Q10.** Minor: Conclusion and abstract should be in one paragraph.

**Questions:**

Please look at the weakness section.

---

> ### Author Response · Authors · 2025-12-02
>
> We sincerely thank Reviewer 3 for their thorough and constructive feedback. We address each point below.
>
> **Strength:**
> We thank the reviewer for acknowledging the core contribution of our work.
>
> **Positional Embeddings and RoPE Compatibility (Q1):**
> As discussed in the general response, positional information is embedded into the key vectors before they are stored in the KV cache, and attention does not re-apply positional encodings during retrieval. Composite tokens operate on these post-embedding representations without modifying positional information, and the attention computation is both permutation-invariant and head-independent. Consequently, reorganization into composite tokens remains compatible with RoPE and standard inference pipelines.
>
> **Streaming Ready Claim (Q2):**
> We agree with the reviewer and thank them for pointing this out. Compared to truly unbounded methods like StreamingLLM, the ``Streaming ready'' column name is misleading, and we will remove it from Table 1. As clarified in the general response, KV cache compression is relevant for streaming inference, but also for scenarios where a compressed cache is computed once and reused across many queries, which is where KVCompose excels. We do not focus on implementing a full streaming system and therefore do not report latency or throughput metrics. KVCompose could still be integrated into frameworks such as LaCache or RocketKV, as outlined in the general response, but we leave this for future work. We will clarify this in the revised manuscript.
>
> **Cache Reorganization Breaking Semantic/Positional Continuity (Q3):**
> We refer the reviewer to our answer to Q1. Since positional information is already embedded in the keys and attention is permutation-invariant over cached positions, composite tokens do not break positional or semantic continuity from the model’s perspective. The compressed cache remains a valid structured tensor that can be stored and reused. Integration with advanced serving optimizations such as speculative decoding would require additional investigation, which we leave to future work.
>
> **Context Length Limitation and Task Diversity (Q4 and Q5):**
> We refer the reviewer to the general response for details on benchmark selection and computational constraints. In brief, Ruler-4096 offers a standardized and diverse evaluation setting widely used in the KV cache compression literature, and the computational cost of exploring larger-scale benchmarks is substantial. Following the reviewers’ recommendation, we add Ruler-16k results, focusing on LLaMA-3.1-8B due to time constraints.
>
> **Missing Comparisons and Computational Overhead (Q6 and Q7):**
> We thank the reviewer for highlighting these methods. ShadowKV is a system for high-throughput long-context inference that exploits the low-rank structure of pre-RoPE keys and relies on custom CUDA kernels for sparse decoding. In contrast, KVCompose is an algorithmic eviction method that operates entirely within standard dense-attention engines without modifying inference runtimes. Our focus is on how to aggregate attention-derived importance scores across heads, layers, and task distributions, and on bridging structured and unstructured eviction via composite tokens. ShadowKV and KVCompose therefore operate at complementary layers of the stack.
> RocketKV is a training-free two-stage compression method for decode-phase acceleration, with a first-stage eviction mechanism and a second-stage hybrid sparse attention process. Importantly, RocketKV is agnostic to the first-stage method, suggesting that KVCompose could replace its coarse-grain compression and potentially improve permanent eviction quality.
> Regarding overhead, as noted in the general response, KVCompose introduces computation during prefill but this is amortized in reusable cache scenarios, which are common in practical deployments.
>
> **Quadratic vs Linear Memory Error (Q8):**
> We thank the reviewer for spotting this error. In the sentence “The cost of the attention mechanism increases linearly with context size during the prefill stage and quadratically in terms of memory required to store KV pairs for each attention head across all layers,” we mistakenly swapped the two terms. As correctly pointed out by the reviewer, the KV cache memory footprint grows linearly with context length, while the computational cost of prefill attention is quadratic. We will correct this in the revised manuscript.
>
> **Q9: FP32 vs FP16/BF16 Precision**
> We thank the reviewer for this suggestion. We will update the memory calculation in Section 2.2 to reflect 2 bytes per parameter, consistent with current deployment practice.
>
> **Q10: Abstract and Conclusion Formatting**
> We thank the reviewer for pointing this out. We have revised the manuscript so that both the abstract and conclusion appear as single paragraphs.

---

### Official Review · Reviewer_Gk8h · 2025-11-01

**Soundness:** 2
**Presentation:** 2
**Contribution:** 2
**Rating:** 4
**Confidence:** 3

**Summary:**

This paper introduces KVCompose, a structured KV cache compression method. It uses attention-guided scoring to identify important tokens , allows each head to select different tokens while maintaining a uniform tensor structure (termed "composite tokens") , and adaptively allocates retention budgets across layers. The method is designed for compatibility with standard inference engines and demonstrates strong performance on the Ruler-4096 benchmark against other structured methods.

**Strengths:**

- The method intelligently allocates a global retention budget across layers based on token importance scores, rather than using fixed heuristics.

- Consistent performance gain is observed across the board.

**Weaknesses:**

- The method requires to collect attention scores, adding complexity.

- Experiments mostly conducted on relatively small models. Would suggest the author conduct experiments on larger models and more challenging problems, like Qwen3-thinking and AIME.

**Questions:**

1. Can you clarify on the concept of "composite tokens"? What's its difference from conventional kv cache? What benefits it brings by viewing kv cache from different layer as a token?

2. The idea for consturcting adaptive kv cache sounds very similar to "Model Tells You What to Discard: Adaptive KV Cache Compression for LLMs".

---

> ### Author Response · Authors · 2025-12-02
>
> **Strength:**
> We sincerely thank the reviewer for recognizing the value of our layer-adaptive budget allocation mechanism and for noting the consistent performance gains across our evaluation. We appreciate the positive feedback on dynamically allocating retention budgets based on composite-token importance scores rather than relying on fixed heuristics.
>
> **Attention Score Collection Complexity (W1):**
> As discussed in the general response, KVCompose relies on aggregated attention scores to estimate token importance, a design principle shared by several performant KV compression methods. While attention is not a perfect indicator in all settings, it provides a practical and efficient proxy, since scores are already computed during the forward pass. The additional complexity introduced by our aggregation strategy is offset by substantial accuracy gains over simpler heuristics. For example, KVCompose improves AUC by +23.7 points over StreamingLLM and +20.8 points over PyramidKV on Ruler-4096, illustrating the benefit of a more faithful importance estimation.
>
> **Model Scale and Benchmark Coverage (W2):**
> We refer the reviewer to the general response for a detailed explanation of our benchmark choices and computational constraints. In brief, Ruler-4096 offers a standardized and diverse evaluation framework widely used in the KV cache compression literature, and the set of models we adopt reflects commonly studied sizes under reproducible research budgets. Following the reviewer’s suggestion, we also report results on Ruler-16k, focusing on LLaMA-3.1-8B due to time constraints.
>
> **Composite Tokens Clarification (Q1):**
> We thank the reviewer for raising this question, and we will incorporate a clarification in the final manuscript. Composite tokens are the key innovation that closes the gap between unstructured and structured eviction methods. In structured compression, all heads must share a uniform sequence dimension, which forces them to retain the same tokens. This constraint limits selectivity and can discard tokens that are critical to specific heads.
> Composite tokens leverage the fact that the sequence length must be shared, but the identity of the retained positions does not need to correspond to the same original tokens across heads. Each head therefore selects its most important tokens independently, and these selections are aligned into shared positions, producing compressed positions that may refer to different original tokens across heads. This design achieves fine-grained, head-level selectivity previously exclusive to unstructured methods requiring custom CUDA kernels, while preserving the uniform tensor layouts that standard inference engines like vLLM and HuggingFace Transformers demand.
> The benefit of composite tokens is evident in our results: KVCompose matches or exceeds SnapKV (which requires per-head variability) despite operating under stricter structural constraints, and remains competitive with KVZip while requiring no engine modifications.
>
> **Relation to "Model Tells You What to Discard" (Q2):**
> We thank the reviewer for highlighting *Model Tells You What to Discard* (Ge et al., 2024), a seminal work that introduced adaptive KV cache compression guided by attention patterns (FastGen). This is indeed an important reference that we will incorporate into our related work discussion.
> FastGen profiles each attention head to classify it into broad behavioral types, such as local-context heads, special-token–focused heads, or globally attending heads, and applies predefined compression policies accordingly. This head-classification strategy established that uniform compression across all heads is suboptimal, a foundational insight for subsequent methods.
> KVCompose operates at a finer granularity. Instead of selecting among fixed per-head policies, we use attention scores to rank individual tokens and introduce composite tokens that allow each head to retain its own most important entries while maintaining a uniform per-layer cache layout and a globally adaptive layer-wise budget. Our systematic study of attention aggregation strategies further complements FastGen's profiling-based approach by examining how importance signals should be combined across heads, layers, and task distributions, design dimensions that prior work treats as fixed but which our ablations show have substantial impact on compression quality.
> In terms of performance, while FastGen established important conceptual foundations, many methods have since advanced adaptive KV cache compression. Our evaluation on the standardized Ruler benchmark demonstrates that KVCompose achieves state-of-the-art results among structured methods.

---

### Official Review · Reviewer_A8M8 · 2025-11-11

**Soundness:** 2
**Presentation:** 2
**Contribution:** 2
**Rating:** 2
**Confidence:** 4

**Summary:**

The paper proposes KVCompose, a structured KV cache compression method for long-context LLM inference, by pre-collecting attention distributions, calculating token importance, constructing "composite tokens," and using global sorting to achieve layer-adaptive retention budget allocation, while maintaining each layer's KV cache still as a uniform length, dense tensor. Token selection is fine-grained at the head dimension, yet implemented as structured eviction.

On long-context benchmarks such as Ruler-4096 and kvpress, compared with methods like TOVA, StreamingLLM, SnapKV, PyramidKV, and DuoAttention, KVCompose can achieve higher compression rates at the same accuracy tolerance and significantly outperforms in the AUC metric.

**Strengths:**

Structured and engineering-friendly

The design of composite tokens is natural and inspiring

Global layer-adaptive budget allocation

The method is conceptually clear and formally complete

**Weaknesses:**

The evaluation benchmarks are narrow and lack real-world tasks and diverse model settings. Currently, the main evaluations focus on Ruler-4096 / kvpress synthetic long-context tasks. These tasks are sensitive to patterns like needle in a haystack / aggregation, but they are insufficient to demonstrate generalizability to real multi-turn dialogues, code, long-form summarization, and RAG scenarios and robustness across more architectures.

Attention = Importance assumption lacks deeper analysis. The method is entirely based on aggregated attention scores, but related literature has pointed out that attention does not always reliably characterize causal contributions. The paper performs an ablation study on pooling methods, but it lacks comparison with gradient/importance estimation, also lacks discussion of failure cases or scenarios where attention is unreliable. Currently, the impression is that it works empirically on this benchmark, but the theoretical and empirical analysis is somewhat shallow.

Semantic and Interpretability Issues of Composite Tokens. Composite tokens combine different original positions from different heads into the same new position, essentially altering the alignment of each head. Although this does not mathematically violate the attention formula, it can make position-wise distribution interpretation difficult. Also, the behavior under positional encoding or certain structured prompt patterns opaque.

The evidence for being practically deployable and user-friendly is insufficient.
The article repeatedly emphasizes that there is no need to customize the kernel and that it can directly integrate with vLLM / HF, which is a selling point; however, it lacks actual end-to-end throughput and latency data;  Memory vs QPS curves under real serving settings;
System-level comparison with existing simple sliding window.

The method relies on a preconstructed task set T to collect attention. If the downstream task distribution changes, do we need to recollect attention and regenerate the mapping? For closed-source or commercial models, how can we efficiently collect enough attention to create a static configuration?

Currently, it is more of a conceptual implementation compatible claim. We would like to see solid metrics supporting the conclusion of being practical & scalable.

**Questions:**

It is recommended to briefly explain in the main text the offline overhead involved in generating compressed configurations.

You can add a diagram for comparison traditional structured eviction, this allows readers to immediately see the structural differences.

In the comparison methods, some, such as SnapKV / KVzip, are reconstruction-based / unstructured, which differs slightly from the goal of this work, which is to only perform structured eviction.

It is recommended to release the implementation to lower the barrier for reproduction.

---

> ### Author Response · Authors · 2025-12-02
>
> **Strengths:** We thank the reviewer for acknowledging the engineering-friendly nature of our approach, the natural and inspiring design of composite tokens, the global layer-adaptive budget allocation mechanism, and the conceptual clarity and formal completeness of our method.
>
> **Narrow benchmarks (W1):**
> We appreciate the reviewer’s concern regarding benchmark diversity. As discussed in the general response, we adopt the Ruler-4096 benchmark from kvpress because it offers a standardized and fair comparison framework, with implementations provided by original authors and coverage across diverse task categories. We agree that broader evaluation is valuable; however, the computational cost of our study is substantial. Following the reviewer’s recommendation, we add results on Ruler-16k to validate performance at longer context lengths, focusing on LLaMA-3.1-8B due to time constraints.
>
> **Attention = Importance assumption (W2):**
> We thank the reviewer for raising this point. KVCompose relies on aggregated attention scores to estimate token importance, a design choice shared by several performant KV compression methods, including SnapKV, H2O, TOVA, and KVzip. While attention does not perfectly capture causal contributions in all settings, it serves as a practical and computationally efficient proxy. The main alternative, gradient-based importance estimation, requires backward passes through the model, rendering it incompatible with inference-time constraints and prohibitively expensive in many settings.
>
> **Semantic and Interpretability Issues / Positional Encoding (W3):**
> We thank the reviewer for raising this point. As detailed in the general response, positional information is already embedded into key vectors before they enter the KV cache, and attention does not re-apply positional encoding during retrieval. Composite tokens operate on these post-embedding representations without altering positional information, and each head attends independently. Therefore, no changes to positional encoding logic are required, and KVCompose remains fully compatible with standard inference pipelines.
>
> **Insufficient deployment evidence (W4):**
> We thank the reviewer for raising this point. KV cache compression is relevant not only for streaming long-context inference but also for scenarios where a compressed cache is computed once and reused across many queries, which is common in practical deployments. As clarified in our general response, the contributions of KVCompose, namely composite tokens with layer-adaptive budget allocation and our systematic study of attention-score aggregation, directly support such use cases. While streaming was not the primary focus of this work, KVCompose could be integrated with frameworks such as LaCache or RocketKV. We will clarify this in the revised manuscript and refer the reviewer to the general answer for additional details.
>
> **Task distribution dependency (W5):**
> This is an important practical concern, and we address it directly by reporting results in two settings: task-aware, where attention is aggregated over representative downstream queries, and task-agnostic, where only the input context itself guides importance estimation. KVCompose outperforms state-of-the-art baselines in both settings.
>
> **Offline overhead explanation (Q1):**
> We thank the reviewer for this suggestion. We will add the following paragraph in the main text explaining the offline overhead involved in generating compressed configurations.
> *“KVCompose introduces computation during prefill: attention aggregation across the task distribution, sorting of per-head importance scores, and indexed gathers to reorganize the cache into composite tokens. This overhead is non-negligible for single-query latency but is amortized in reusable cache scenarios, where compression occurs once and the resulting cache serves many subsequent queries.”*
>
> **Diagram comparing eviction types (Q2):**
> We thank the reviewer for this suggestion. We will add a diagram comparing traditional structured eviction, unstructured eviction, and our composite token approach, allowing readers to immediately see the structural differences.
>
> **Comparison with unstructured methods (Q3):**
> We agree that our primary focus is structured eviction. As clarified in the general response, we also report comparisons to unstructured methods such as KVzip and Ada-variants for transparency, since they operate under fewer constraints and provide an upper bound on achievable accuracy. KVCompose matches or exceeds SnapKV under stricter structural requirements, and remains competitive with KVzip without requiring custom kernels, supporting our claim that composite tokens narrow the gap between structured and unstructured eviction while preserving deployability.
>
> **Code release (Q4):**
> We agree with the reviewer and thank them for this suggestion. We are waiting for the conclusion of the review process due to double-blind requirements to release the code publicly.

---

### Author Response · Authors · 2025-12-02
**General Response to Common Reviewer Concerns (2)**

## **Attention-Based Importance Scoring: Rationale and Alternatives**

KVCompose relies on aggregated attention scores to estimate token importance, a design choice shared by several performant KV compression methods including SnapKV, H2O, TOVA, or KVZip. While attention is not perfect, it serves as a practical and computationally efficient proxy: attention scores are already computed during the forward pass, so aggregating them incurs marginal overhead. The principal alternative, gradient-based importance estimation, requires backward passes through the model.

## **Benchmark Scope, Diversity, and Computational Cost**

Our evaluation centers on the **Ruler-4096** benchmark from *kvpress*, which provides a standardized framework where competing methods are typically implemented by their original authors, ensuring fair and reproducible comparisons. Ruler covers diverse task categories, needle-in-haystack retrieval, multi-key and multi-value extraction, aggregation, variable tracking, and question answering, offering breadth beyond any single synthetic pattern. We recognize that evaluation on longer contexts and additional benchmarks would be valuable. However, the computational cost of our study is non-negligible: each configuration requires approximately 4 hours on high-end GPUs, and with 9 compression rates, 7 methods, 3 models, and 2 settings (task-aware and task-agnostic), our experiments total over 1,500 GPU-hours, excluding debugging and exploratory runs. Following reviewer recommendations, we extend our evaluation to **Ruler-16k** to validate performance at longer context lengths; due to time constraints, these additional experiments focus on LLaMA-3.1-8B.

### **Ruler-16k Results (LLaMA-3.1-8B)**
| Method            | 0%   | 50%  | 75%  | AUC  |
|-------------------|------|------|------|------|
| KVCompose         | **95.3** | 87.6 | **70.9** | **74.4** |
| ExpectedAttention | **95.3** | 86.0 | 66.4 | 72.7 |
| TOVA              | **95.3** | 77.9 | 68.4 | 70.1 |
| Duo               | **95.3** | **90.1** | 59.1 | 72.4 |
| SnapKV            | **95.3** | 68.0 | 43.1 | 60.1 |

## **Practical Deployment: Overhead and Reusable Cache Scenarios**

KVCompose introduces computation during prefill: attention aggregation across the task distribution, sorting of per-head importance scores, and indexed gathers to reorganize the cache into composite tokens. This overhead is non-negligible for single-query latency but is amortized in **reusable cache scenarios**, where compression occurs once and the resulting cache serves many subsequent queries. Such scenarios are common in practice: enterprise documentation systems, product-manual chatbots, and coding assistants all operate over fixed reference contexts that persist across user sessions. For these workloads, the one-time compression cost is offset by repeated inference savings, and the accuracy gains of KVCompose over simpler heuristics translate directly into serving quality. Besides, we note that most competing methods in the *kvpress* benchmark are evaluated on HuggingFace implementations rather than production-optimized engines like vLLM; this is standard practice, as modifying vLLM's internals requires significant engineering effort. Importantly, because KVCompose maintains a structured cache with uniform tensor layouts, the compressed cache can be used directly with standard dense attention: no custom CUDA kernels are required. This stands in contrast to unstructured methods, which would need specialized sparse-attention kernels to translate their per-head variability into actual runtime gains, a capability that existing sparse-matrix primitives do not yet reliably provide.

## **Task-Aware vs. Task-Agnostic Compression**

A practical concern for deployment is whether KVCompose requires knowledge of downstream tasks to construct effective compression configurations. We address this directly by reporting results in two settings: **task-aware**, where attention is aggregated over representative downstream queries, and **task-agnostic**, where only the input context itself guides importance estimation. KVCompose outperforms state-of-the-art baselines in both settings. The task-agnostic configuration is particularly relevant for closed-source or commercial deployments where downstream query distributions may be unknown or shifting; it demonstrates that our method does not depend on privileged access to task-specific data and remains robust under distribution uncertainty.

---

### Author Response · Authors · 2025-12-02
**General Response to Common Reviewer Concerns (1)**

## **KVCompose Contributions**

This work makes two principal contributions to the KV cache compression literature. First, we introduce the notion of **composite tokens**, which closes the gap between unstructured and structured eviction methods by allowing each attention head to independently retain its most important tokens while aligning them into shared slots. Our approach achieves fine-grained, head-level selectivity, previously exclusive to unstructured methods requiring custom CUDA kernels, while preserving the uniform tensor layouts that standard inference engines like vLLM and HuggingFace Transformers demand. Composite tokens are complemented by a **layer-adaptive budget allocation mechanism**: rather than prescribing fixed per-layer schedules, we allocate a global retention budget across layers by ranking composite-token scores, ensuring that layers carrying more informative tokens receive larger capacity and improving robustness under aggressive compression. Second, we provide a systematic study of how to aggregate attention-derived importance scores across heads, layers, and task distributions, offering design insights that apply broadly to the family of attention-based KV compression methods.

## **Structured vs. Unstructured Methods: Why We Compare to Both**

Our primary focus is **structured eviction**, methods that maintain uniform tensor layouts compatible with existing inference engines. However, we also report results for unstructured approaches such as KVzip and Ada-variants, which allow per-head budget variability and achieve higher flexibility at the cost of requiring custom CUDA kernels. We include these comparisons for transparency: unstructured methods represent a less constrained problem and thus provide an upper bound on achievable accuracy. Notably, KVCompose matches or exceeds SnapKV despite operating under stricter structural constraints, and remains competitive with KVzip while requiring no engine modifications. This validates the central claim of our work: composite tokens effectively bridge the gap between structured and unstructured eviction, capturing much of the latter's flexibility without sacrificing deployability.

## **KVCompose and Streaming Settings**

KV cache compression is central to streaming long-context inference, improving throughput and enabling longer feasible contexts, but it also supports regimes where a compressed cache is computed once and reused across many queries. Pertinent examples include enterprise knowledge bases where internal documentation is preloaded for employee Q&A, customer support chatbots operating over fixed product manuals or policy documents, and coding assistants equipped with library documentation or codebase overviews to provide contextually grounded suggestions. In all such settings, aggressive one-time compression with minimal quality loss, precisely the regime where KVCompose excels, translates directly into serving efficiency. Our focus is on (i) how to aggregate attention-derived importance scores across heads, layers, and task distributions, and (ii) how to bridge structured and unstructured eviction via composite tokens that retain per-head selectivity while preserving a uniform cache layout. Streaming was not the primary focus of this work; however, KVCompose could be integrated with streaming frameworks such as **LaCache (D. Shi et al. 2025)**’s iterative compactification or **RocketKV (P. Behnam et al. 2025)**’s two-stage pipeline. In particular, RocketKV's architecture is agnostic to the first-stage eviction mechanism, suggesting that KVCompose could serve as a drop-in replacement for coarse-grain compression, with our attention-aggregation insights informing both permanent eviction and subsequent dynamic token selection. We leave this for future work.

## **Composite Tokens and Positional Encoding**

A natural concern with composite tokens is their interaction with positional encodings, particularly Rotary Positional Embeddings (RoPE) used by modern LLMs such as LLaMA and Qwen. We clarify that positional information is embedded into the **key vectors before they are stored** in the KV cache; this is how both HuggingFace Transformers and production serving engines like vLLM operate, including under PagedAttention and FlashAttention. Once cached, attention computation does not re-apply positional encoding; it simply retrieves the precomputed keys and values. Moreover, the attention operation itself is permutation-invariant over cached positions and head-independent: each head attends to its own keys without cross-head coordination. Composite tokens exploit precisely this structure: they reorganize post-embedding key-value pairs across heads, never altering the positional information already encoded within each vector. Consequently, no modifications to positional encoding logic are required, and the approach remains compatible with standard inference pipelines without engine changes.

---

### Meta-Review · Area_Chair_QUYi · 2026-01-01

**Summary:**

The reviewers raised concerns mainly on evaluation scope, practical deployability evidence, and conceptual clarity. While all reviewers acknowledged that the paper introduces a new and well-motivated idea, the main concern lies on whether the empirical evaluation sufficiently demonstrates the real-world applicability.

Specifically, main questions include 1) the evaluation on some synthetic long-content benchmarks (Rule-4096) rather than real dialogue and code scenarios. 2) the assumption that aggregated attention reflects token importances with limited analysis of failure modes or alternatives; 3) lack of system-level metrics such as throughout, latency or memory-QPS tradeoffs.

The rebuttal clarified many of the questions such as the motivation, theoretical validity, etc. After rebuttal, I think the remaining concerns are largely about breadth and depth of evaluations.

**Reviewer Concerns:**

Concerns addressed:
1. The confusion of how the composite tokens differ from conventional structured eviction and how they interact with positional encodings was addressed.
2. The concerns that the method requires knowledge of downstream tasks was mitigated by extra task-agnostic results, demonstrating the method performs well even when importance estimation is derived solely from the context.
3. The rebuttal explained the method targets reusable-cache scenarios where one-time compression cost is amortized.
4. There are additional results on evaluation at longer context lengths, even though it is limited to one model.

Concerns partially addressed or remained:
1. While there are additional results for long context lengths, more results on real scenarios such as dialogue, code, summarization and RAG are recommended.
2. The rebuttal provides justifications on the attention-as-importance assumption, but does not deep dive on the failure cases or compare against gradient-based or causal importance measures.
3. Direct measurement on system-level deployment such as latency and throughput are still lacking.

**Reviewer Scores:**

Based on the analysis, I expect Reviewers A8M8 and ZsMN might raise their scores but would still under reject, as some concerns about benchmark realism might still remain. The other reviewer might raise the scores to borderline. But given the negative scores, I think they would not reach a consensus of accept.

---

### Decision · Program_Chairs · 2026-01-26

Reject